# Explainability of deep reinforcement learning algorithms in robotic domains by using Layer-wise Relevance Propagation

## Abstract

A key component to the recent success of reinforcement learning is the introduction of neural networks for representation learning. Doing so allows for solving challenging problems in several domains, one of which is robotics. However, a major criticism of deep reinforcement learning (DRL) algorithms is their lack of explainability and interpretability. This problem is even exacerbated in robotics as they oftentimes cohabitate space with humans, making it imperative to be able to reason about their behaviour. In this paper, we propose to analyze the learned representation in a robotic setting by utilizing graph neural networks. Using the graphical neural networks and Layer-wise Relevance Propagation (LRP), we represent the observations as an entity-relationship to allow us to interpret the learned policy. We evaluate our approach in two environments in MuJoCo. These two environments were delicately designed to effectively measure the value of knowledge gained by our approach to analyzing learned representations. This approach allows us to analyze not only how different parts of the observation space contribute to the decision-making process but also differentiate between policies and their differences in performance. This difference in performance also allows for reasoning about the agent's recovery from faults. These insights are key contributions to explainable deep reinforcement learning in robotic settings.

## 1 Introduction

While Deep Reinforcement Learning (DRL) has shown tremendous success in domains like games, highly structured robotic settings, and other real-world domains, it is still held back by concerns over its safety and explainability. Due to DRL leveraging non-linear function approximators (i.e., neural networks), its behaviour cannot be fully understood and anticipated. Especially in domains where DRL is deployed alongside humans, it is expected to perform as anticipated by those humans. To fully harness the potential that comes from this powerful technique, it is paramount to translate the internal state of DRL approaches into human-understandable signals.

A Reinforcement Learning (RL) agent interacts with the environment to gain knowledge and learn to perform. It observes the environment, takes action accordingly, receives feedback, and updates its behavior based on the feedback. This self-training ability makes RL a complex learning procedure, causing many challenges in interpreting its behavior. Combining RL with the representation learning power of Deep Learning (DL) models further adds to this complexity. Explaining the policy learned by a black box DL model as a function approximator is one of the major challenges in interpreting DRL models. One method proposed to tackle this challenge is State Representation Learning (SRL). SRL is a feature learning method that learns a low-dimensional representation of the state from high-dimensional raw observations (like pixels of an image) by capturing the variation in the environment caused by the agent's actions. (Lesort et al., 2018; Doncieux et al., 2018; Raffin et al., 2018; 2019; Traoré et al., 2019; Doncieux et al., 2020). While SRL methods identify the most relevant features of a high-dimensional observation for learning to act and compact the observation accordingly, we still require highlighting the most relevant features in low-dimensional compact observation space robotic environments.

This work aims at identifying the relevance of each entity of a robot in the decision-making process in low-dimensional sensory input robotic environments in which observation space is as compact as possible. Compact observation means that removing any part of the observation space would lead to a drop in performance. Saliency methods have proved to be successful in highlighting the most relevant pixels in image classification (Simonyan et al., 2013; Bach et al., 2015; Zhou et al., 2016; Selvaraju et al., 2017; Zhang et al., 2018a;b), and entities and relations in graph classification (Baldassarre & Azizpour, 2019; Pope et al., 2019). Some work extended the application of saliency methods from classification to RL, focusing on environments with visual data as states. One example of this application is explaining the DRL agent's behavior in Atari games by visualizing its decisions (Weitkamp et al., 2018; Greydanus et al., 2018; Iyer et al., 2018; Huber et al., 2019). Nevertheless, saliency methods in RL have only been applied to RL problems with visual input states. In our work, a saliency method is used to highlight the contribution of each part of the robot to the policy, which helps us identify the most and least contributing parts to the decision-making.

Since the structure of a robot is similar to graphs, we represent the robot's state using graphs and apply a saliency technique to highlight the contribution of each part of the graph to the agent's decisions. Baldassarre & Azizpour (2019) claim that Layer-wise Relevance Propagation (LRP) proves to be efficient in identifying the most contributing parts of a graph to a graph classification task. Based on this claim, we choose LRP as our saliency method. First, we need to use graph neural networks (GNN) as function approximators in our DRL algorithm. After the agent's performance converges, we apply LRP to identify the most contributing components of the robot to learning the task.

A robot contains some number of body parts that are connected through joints. The body and joint in the robot correspond to the node and edge in the graph, respectively. Graph representation is used to decompose the robot into its entities and their relationships. This kind of representation in DRL has been previously used by Sanchez-Gonzalez et al. (2018) and Wang et al. (2018). The LRP highlights the relevance of every action element in the output to each entity of the observation graph, creating a heat map of action-entity relevance, based on which we can distinguish the most relevant parts from the least relevant ones.

Knowing the contribution of every entity in the robot to the decision-making process is highly important. One application is to provide a **visualization for explaining the training process**, which can be done by identifying the robot's entities contributing to learning a task during the training process. To get an intuition, assume a child is learning to stand up. During the initial stages of learning, they use their hands as assistance; however, in later stages, they can stand up easily without using their hands. Therefore, during the early stages of training, the contribution score of both hands and legs would be high, while during later stages, the contribution of hands drops. Another application is during a malfunction, where part of a robot is broken. Knowing the importance of the broken part helps us **figure out how severe the damage is** and whether the agent can recover from that malfunction or not. This recovery can be in the form of learning a new policy from scratch for the new dynamics or transferring the policy trained in the previous dynamics. If we choose to adapt to the new dynamics after a malfunction, this method can **explain the adaptation process**. To have better intuition, imagine the human's writing task. A right-handed human breaks their right hand; after that, they start using their left hand instead. In the first dynamics, the contribution of their right hand is the highest in the writing task. In contrast, in the second one, after adaptation to the new dynamics, the importance of their left hand escalates while the right hand's importance drops.

## 2 BACKGROUND

### 2.1 GRAPH NEURAL NETWORK

*Graphs* are tools that are used to represent structured data. A graph contains multiple entities with relationships among them. Entities and their relationships are shown using nodes and edges of the graph, respectively, which gives us flexibility in designing representation architectures of arbitrary shapes. Furthermore, this way of knowledge representation emphasizes the location of each entity relative to other entities. *Graph neural networks* (GNN) are neural networks that operate on graph inputs. GNNs impose constraints on relationships and interactions among entities while finding the optimal solution. In other words, they emphasize the *relational inductive bias*. Our GNN architecture and operations are according to Battaglia et al. (2018). In an input graph, there are three kinds

of features: 1) *node feature* which shows the state of the entity, 2) *edge feature* which illustrates the state of each relation, and 3) *global feature* which indicates the state of the whole system. $E = \{e_k\}$ and $V = \{v_i\}$ denote the set of edge and node feature vectors respectively and $u$ denotes the global-level feature vector. A GNN has three operations for updating node, edge, and global-level features. Equation 1 shows the operations in a GNN.

$$
\begin{aligned}
\mathbf{e}'_k &= \phi^e(\mathbf{e}_k, \mathbf{v}_{r_k}, \mathbf{v}_{s_k}, \mathbf{u}) & \overline{\mathbf{e}}'_i &= \rho^{e \to v}(E'_i) \\
\mathbf{v}'_i &= \phi^v(\overline{\mathbf{e}}'_i, \mathbf{v}_i, \mathbf{u}) & \overline{\mathbf{e}}' &= \rho^{e \to u}(E') \\
\mathbf{u}' &= \phi^u(\overline{\mathbf{e}}', \overline{\mathbf{v}}', \mathbf{u}) & \overline{\mathbf{v}}' &= \rho^{v \to u}(V')
\end{aligned}
\tag{1}
$$

where $E'_i = \{(\mathbf{e}'_{\mathbf{k}}, r_k, s_k)\}_{r_k=i,k=1:N^e}$ is the updated set of edges whose receiver is node $i$ and $r_k$ and $s_k$ are the receiver and sender nodes of edge $k$, $V' = \{\mathbf{v}'_{\mathbf{i}}\}_{i=1}^{N^v}$ is the set of updated nodes, $E' = \bigcup_i E'_i = \{(\mathbf{e}'_{\mathbf{k}}, r_k, s_k)\}_{k=1}^{N^e}$ is the set of all the updated edges. $\phi^e$ updates edge features incorporating surrounding nodes and global-level features in the update. $\phi^v$ updates node features incorporating aggregated features of the surrounding edges of a node and global-level features. Finally, $\phi^u$ updates global-level features given the graph's aggregated node and edge features (Battaglia et al., 2018). A summary of the GNN operations can be seen in Appendix A.

## 2.2 LAYER-WISE RELEVANCE PROPAGATION

Originally, the LRP has been used as a visualization tool highlighting the contribution of pixels to the classification of images by neural networks. The LRP decomposes the output probability given to a specific class and back-propagates this score to the input components so that the sum of scores in each layer is equal across layers:

$$
f(x) = \cdots = \sum_{d \in (l+1)} R_d^{l+1} = \sum_{d \in l} R_d^{(l)} = \cdots = \sum_d R_d^{(1)}
\tag{2}
$$

where $R_d^l$ is the relevance score given to unit $d$ of layer $l$ and $f(x)$ is the score of the output layer. Bach et al. (2015) proposed two methods for calculating these relevance scores and preventing them from taking unbounded values. These two methods are the $\alpha\beta$-rule and the $\varepsilon$-stabilized rule. As discussed by Baldassarre & Azizpour (2019), the latter rule is more robust and simple, which is why we also use the second rule. For the details about LRP, please refer to Appendix B.

Intuitively, during the forward pass, the neural network emphasizes some parts of the input that are contributing more to the output by giving them higher weights and activating them. LRP uses these weights and activations in each layer to propagate the output back through the network until the input layer. Hence, it can highlight the most contributing parts of the input.

## 2.3 DEEP REINFORCEMENT LEARNING

Our method can explain all the policies with neural networks as function approximators regardless of the type of DRL algorithm. We selected Soft Actor-Critic (SAC), popular and known to perform well in robotic environments (Haarnoja et al., 2018a). Our problem is a policy search in a Markov decision process defined by a tuple $(\mathcal{S}, \mathcal{A}, p, r)$ where $\mathcal{S}$ and $\mathcal{A}$ are continuous state and action spaces, $p : \mathcal{S} \times \mathcal{S} \times \mathcal{A} \to [0, \infty)$ is the state transition probability density, and $r : \mathcal{S} \times \mathcal{A} \to [r_{\min}, r_{\max}]$ is the environment's reward function. The policy, denoted by $\pi$, is the probability of selecting action $a_t$ in state $s_t$ at time step $t$. SAC is trained through an off-policy process, meaning that the policy being evaluated and updated is separate from the policy used for generating trajectories. SAC's objective function maximizes not only the expected cumulative reward but also an entropy term. Therefore, the optimal policy looks as follows:

$$
\pi^* = \arg\max_{\pi} \sum_t \mathbb{E}_{(s_t, a_t) \sim \rho_\pi} \left[ r(s_t, a_t) + \alpha \mathcal{H}(\pi(.|s_t)) \right]
\tag{3}
$$

where $\alpha$ is the temperature parameter that determines the importance of entropy, and $\mathcal{H}(\pi(.|s_t))$ is the entropy term. This entropy term facilitates the optimization process by smoothing out the objective function to avoid getting stuck in local optima. Empirically, this entropy term incentivizes the agent to explore the environment more widely. For the details, please refer to the original paper (Haarnoja et al., 2018b). We also use automatic temperature parameter tuning, as discussed in the paper.

## 3 PROPOSED METHOD

Our method has two phases. In the first phase, we train the agent until convergence using graphs as input observations and GNNs as function approximators for the DRL algorithm. Afterward, we apply the LRP algorithm to the learned policy to provide a heatmap showing the relevance between each action element and observation entity across time-steps. Then, by using the LRP scores, we identify the most critical entities and joints in the observation and action spaces, respectively. The importance scores calculated in this phase are evaluated in the second phase. Each joint plays two roles in decision-making. First, its features are given to the agent as the state. Second, an action in the form of torque is applied to that joint, creating motion. Therefore, there are two kinds of information extracted by the LRP:

1. **Entity Importance in the Observation Space:** Each action element gives a relevance score to each entity of the observation space. These relevance scores are averaged for an entity across actions to yield the importance of that entity to the whole decision-making process.

2. **Joint Importance in the Action Space:** The relevance scores given by each action to entities of the observation space are averaged for each action element (i.e., joints) across entities to yield the importance of the joint in the action space.

Note that we use the terms "Entity" and "Joint" to denote the observation and action space elements, respectively. The reason is that not all the elements of the observation space are "Joints", unlike the elements of the action space. For each environment, the evaluation targets the two kinds of information provided above. To evaluate the importance of the entities in the observation space, we **occlude** the features of that entity by removing its features from the observation space and then rerun the experiments in the new (partially observable) environment. We can validate the importance score for each entity based on the drop in performance, which is expected to be proportional to the importance score given to the occluded entity. For the importance of a joint in the action space, we **block** that joint so that no torque (motion) can be applied to it. Again, we can validate the importance scores for each joint based on the amount of drop in the performance, which is expected to be proportional to the importance score given to the blocked joint.

A high-level sketch of the problem is to identify the contribution of each component of the robot to the decision-making process. The first step of our solution is to treat each part as an entity with relationships between each pair of entities, then learn a policy on the environment using this decomposition. Hence, the state $s_t$ at time step $t$ would be in the form of:

$$s_t = G(V_t, E_t, u_t) \qquad (4)$$

where $V_t$ and $E_t$ are the set of node and edge features at time step $t$ respectively and $u_t$ is the global-level feature vector. To use graph representation by the DRL, we need to replace fully-connected networks as function approximators with GNNs. There are 5 networks in SAC as follows: policy network $\psi$, Q-networks $\theta_1$ and $\theta_2$, and target networks $\bar{\theta}_1$ and $\bar{\theta}_2$ as explained in the algorithm (Haarnoja et al., 2018b). Wherever we have states as input, we use a GNN architecture to extract features from the state.

Within one layer of a GNN, three operations are done in order as in equation 1: 1) update edge features using $\phi^e$, 2) update node features using $\phi^v$, 3) and update global features using $\phi^u$. In our architecture, for $\phi^e$, we avoid incorporating node or global features in the update; for $\phi^v$, we avoid incorporating global features in the update (see equation 1). This avoidance helps us speed up the network update without any drop in performance. However, for $\phi^u$, the update is the same as mentioned in equation 1. The first and second GNN layers update edge, node, and global feature vectors' sizes to 256 and 128, respectively. All the aggregation functions $\rho$ are average pooling. The global feature vector of the output graph is the network's output. This output is concatenated with the action for the Q-networks and fed into a 3-layer fully-connected network of size 256. For the policy network, since there are two output layers: one for the *mean* and another for the *standard deviation* of the Gaussian distribution, we use different $\phi^u$ functions at the output layer for each one of them.

After training the agent until convergence, we proceed with the explanation phase. In this phase, there is no further update to the policy, and we want to analyze the learned behavior. At each

time-step $t$, given the state of the environment, the policy network outputs the action $a_t = \pi_\psi(s_t)$ corresponding to state $s_t$. This action, which is the mean of the Gaussian distribution, would look like the following:

$$a_t = [a_t^{(1)}, a_t^{(2)}, \ldots, a_t^{(h)}] \tag{5}$$

where $h$ is the number of action elements corresponding to the number of actuated joints. To calculate the relevance of each action $a_t^{(i)}$ where $i \in \{0, \ldots, h-1\}$, to the input graph components, we zero out all the elements in the action vector except the element at index $i$, which forms the relevance score of the action $i$ at time step $t$, $r_t^{(i)}$ in the output graph's global features. If $e_i \in \mathbb{R}^h$ denotes one hot vector whose elements are zero except the one at index $i$ (which equals 1), then

$$r_t^{(i)} = a_t \cdot e_i \tag{6}$$

This relevance score is set to the global features of the output graph, then back-propagated to the input. If we denote the layer-wise relevance propagation operation on a neural network with $LRP()$,

$$R_{c,t}^{(i)} = LRP(r_t^{(i)}) \quad \text{for } c \text{ in } \{0, \ldots, C-1\} \tag{7}$$

where $C$ is the number of components of the input graph, $R_{c,t}^{(i)}$ is the relevance of action $i$ to component $c$ of the input graph at time-step $t$. The $LRP()$ back-propagates the vector $r_t^{(i)}$ to the input graph's components. Then, the relevance of each action to a corresponding component of the input is averaged across time steps. The evaluation phase is summarized in algorithm 1.

---

**Algorithm 1** Calculating relevance scores for the components in the observation space.

---

1: **let** $R_c^{(i)} = 0$ for $c \in \{0, \ldots, C-1\}$ and $i \in \{0, \ldots, h-1\}$ ▷ $R_c^{(i)}$ is the relevance score given by action $i$ to component $c$ of the input. $c$ can be either a node, an edge, or the global unit
2: **let** $N$ denote the number of episodes
3: **for** each episode $n$ in $\{0, \ldots, N-1\}$ **do**
4:     **for** each time-step $t$ in $\{0, \ldots, T-1\}$ **do**
5:         Sample current state $s_t$ of the environment
6:         $a_t = \pi_\phi(s_t)$                                  ▷ $a_t$ equals the mean of the policy distribution
7:         **for** each element $i$ of the action vector **do**
8:             $r_t^{(i)} = a_t \cdot e_i$
9:             $R_c^{(i)} = R_c^{(i)} + LRP(r_t^{(i)})$
10:         **end for**
11:     **end for**
12: **end for**
13: **return** $\forall c, i : R_c^{(i)}/(T \times N)$                  ▷ Average of the relevance across time-steps

---

## 4 EXPERIMENTS

The experiments are run across two simulated robotic environments in MuJoCo Todorov et al. (2012) OpenAI Gym OpenAI (2021): `Walker2D-v2` and `FetchReach-v1`. The observation space is converted to a graph with only edge and global features. The walker is a two-dimensional two-legged figure that consists of four main body parts – a single torso at the top (with the two legs splitting after the torso), two thighs in the middle below the torso, two legs in the bottom below the thighs, and two feet attached to the legs on which the entire body rests. The goal is to coordinate the torso to move forward by applying torques on the six joints connecting the six body parts OpenAI (2022). The fetch robot is a robot arm consisting of 7 joints ordered in a line, where the `shoulder_pan` is the joint closest to the base, and the `wrist_roll` is located at the other end, connected to the end-effector. The goal is to move the end-effector to a specific 3D position given by the environment, i.e., the `goal`. The actions for both environments are torques applied to joints. For both `Walker2D-v2` and `FetchReach-v1`, the edge feature is the position and velocity of the joint. In `Walker2D-v2`, the `torso` in the model is a joint connecting the robot to the world, showing its position and velocity. In `FetchReach-v1`, the global feature is the `goal`'s position.

### 4.1 TRAIN AND EXPLANATION PHASE

This phase starts by training the DRL agent until convergence on the two environments, followed by applying LRP to the learned policy to generate a heat map. This heat map represents the relevance of each action element to each observation entity, as in Figure 1. The relevance scores for each environment are calculated and averaged across 10 seeds, and each run for 20 episodes. We can calculate contribution scores for each entity and joint by averaging LRP scores across every observation entity and action element separately [1]. The *entity importance* score is on the top-left and the *action importance* score is on the bottom-left of Figures 2 and 3 for `Walker2D-v2` and `FetchReach-v1`, respectively. The correctness and intuition of the following hypotheses are discussed in Section 4.2.

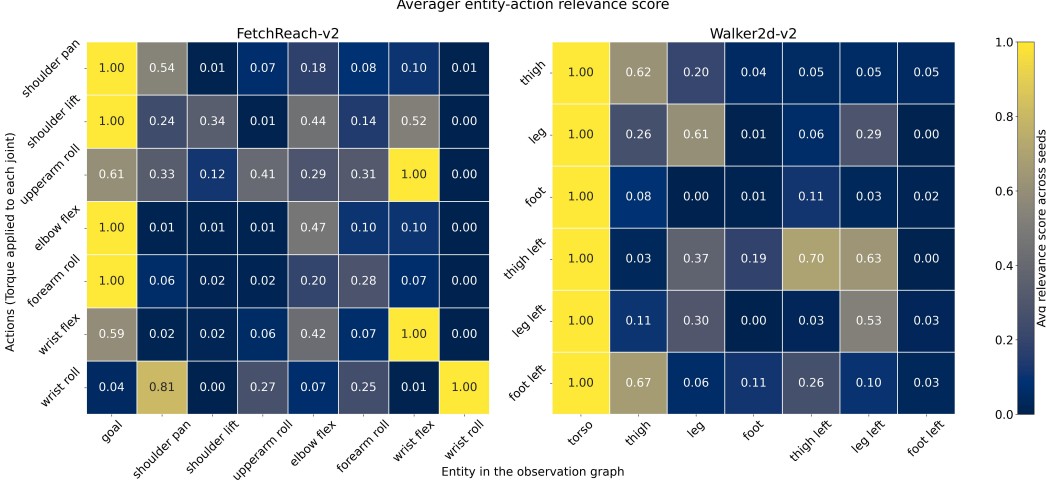

Figure 1: The LRP heat map for action-entity relevance score. The y-axis and x-axis show elements of the action and entities of the observation, respectively. Since joints have different characteristics and possible amounts of torque, the actions have different ranges. Therefore, we normalize the relevance scores for each action across all the observation entities by dividing by the maximum score given by that action.

#### 4.1.1 WALKER2D-V2

Figure 1 right represents the heat map produced for the `Walker2D-v2` environment by the LRP. As the heatmap indicates, scores given to the `torso` entity are the highest across all the elements of the action space. The reason is that the goal of the environment is to increase the speed of the torso. Another thing that we expect to observe is the high relevance of the action elements to their corresponding joint entities in the observation space. By ignoring the `torso` column from the heat map, this dependence can be nicely seen on the heat map in which the relevance scores around the diagonal are relatively high. For example, for the action (torque) applied to the `thigh` joint, the relevance score given to the `thigh` entity (i.e., 0.62) is high relative to other entities. The same is true for `leg`, `thigh_left`, and `leg_left`. For the `thigh_left` action, in addition to `thigh_left` entity, the relevance score given to the `leg_left` entity is also high. This case highlights one of the reasons we selected the graph structure: not only do we take into account the effect of features of each entity on the decision-making process, but we also consider their position in the structure. In other words, the vicinity of the two entities in the robot is the cause of the high score given to the `leg_left` entity by `thigh_left` action.

According to the *entity importance* plot (top-left bar plot in Figure 2), the most important entity to the policy is `torso` – the goal of the environment. The importance score for other entities is pretty small; however, the score of `leg`, `thigh_left`, and `leg_left` are higher than the remaining.

---

[1]The importance scores are not the average scores on the heat map because the scores on the heat map are normalized for each action.

Moreover, the *action importance* plot (bottom-left bar plot in Figure 2 indicates that the critical action elements are `foot` and `foot_left` joints, respectively.

### 4.1.2 FETCHREACH-V1

Figure 1 left shows the heat map generated by the LRP for the `FetchReach-v1` environment. The position of the `goal` receives a high score across all actions (except `wrist_roll` ). This is because all the torques should be adjusted in a way to position the end-effector at the `goal` . Ignoring the `goal` column, similar to `Walker2D-v2` 's heat map, we can see a diagonal pattern. The highest scores for `shoulder_pan` , `elbow_flex` , `forearm_roll` , `wrist_flex` , and `wrist_roll` actions belong to their corresponding joint entities in the observation space as expected. Although for `shoulder_lift` and `upperarm_roll` actions the scores on the diagonal are unexpectedly not the highest, still the scores given to their corresponding joint entities in the observation space are relatively high. This unexpected result might be explained using the graph structure and the vicinity of joints: for `shoulder_lift` and `upperarm_roll` actions, the relevance scores are distributed across multiple entities in the observation space.

According to the *entity importance* plot (top-left bar plot in Figure 3), the most important entity to the policy is `goal` . Other than `goal` , `shoulder_pan` , `forearm_roll` , `wrist_roll` , and `upperarm_roll` have relatively high importance respectively. Other entities have pretty small importance scores. Moreover, the *action importance* plot (bottom-left bar plot in Figure 3) indicates that the critical joints for positioning the robot's end-effector to the `goal` are `elbow_flex` , `wrist_flex` , and `forearm_roll` joints, respectively.

### 4.2 EXPLANATION EVALUATION PHASE

The purpose of this section is to validate the correctness of the hypotheses mentioned in Section 4.1. To ensure that the contribution of each part to the decision-making does not depend on the type of neural network, we use fully-connected networks in this phase as function approximators. As discussed, the evaluation targets the importance of entities in the observation and joints in the action. The first one is fulfilled by **occluding** the entity in the observation space, and the second by **blocking** the joint. In each case, their importance scores are validated based on the drop in performance. In Figures 2 and 3, the upper and lower rows reflect the importance of the observation entities and the action elements evaluation results, respectively. The importance scores are normalized by dividing by the maximum score and the performance bars are divided by the performance of the `standard` setting. The `standard` setting is the original environment with no occlusion in entities nor block in joints. The significance tests are t-test with 95% confidence interval, meaning that the two performances are significantly different for $p \leq 0.05$. Also, the results for performance bars of `Walker2D-v2` and `FetchReach-v1` are averaged across 30 and 10 seeds, respectively. The reason is that `Walker2D-v2` 's performance was unstable across seeds with high variance, while `FetchReach-v1` was stable with a low variance.

### 4.2.1 WALKER2D-V2

First, we discuss the observation entity importance evaluation in the upper-row plots of Figure 2. The top-left bar chart shows that the most important entity in the observation space is the `torso` , with a score of 1.0 . The importance bars for entities other than `torso` are lower than or equal to 0.3 , among which `leg` , `thigh_left` , and `leg_left` have the highest scores. By occluding the `torso` from the observation space, the agent could not learn a policy at all. Therefore, it has the most significant drop in performance compared to the `standard` case in the upper-middle plot of Figure 2. The performances of the `thigh` , `leg` , `thigh_left` , and `leg_left` are equal, as can be seen in their performance bars and significancy test (i.e., their performances are not significantly different) on the upper-middle and -right of the Figure 2. The `foot` and the `foot_left` have the least drop in performance compared to the `standard` setting, as shown in their performance bar, which can be deduced from their importance score. Nevertheless, the importance score for the `thigh` is unexpected because, based on its performance bar, we expect its score to be noticeable, similar to `leg` , `thigh_left` , and `leg_left` .

The joint importance in the action space evaluation is reflected in the lower-row plots of Figure 2. As the bottom-left bar chart indicates, `foot` and `foot_left` joints are the most important. Their

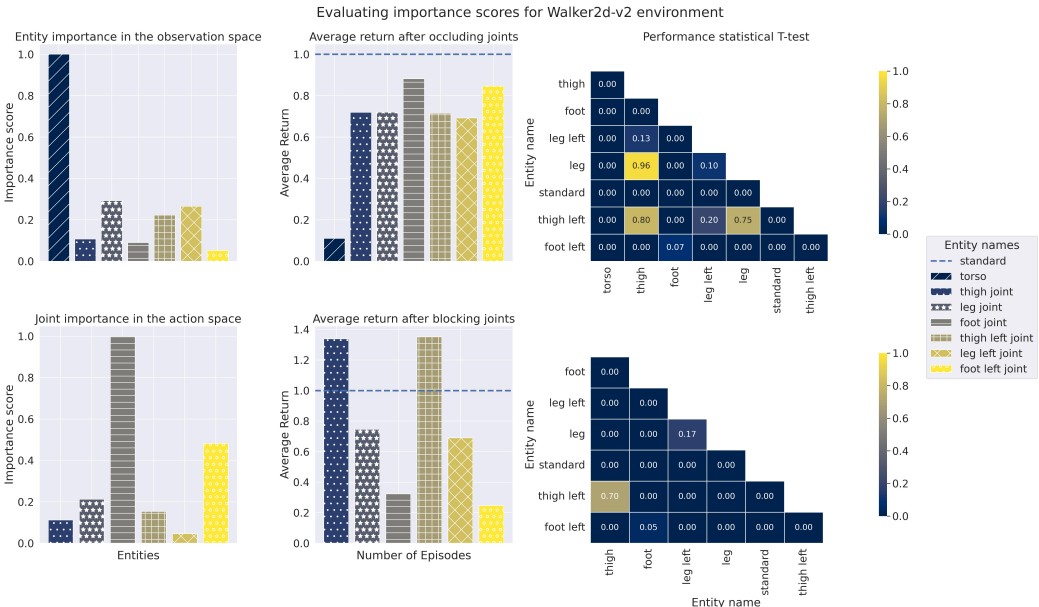

Figure 2: Evaluating explanation for the `Walker2D-v2`. Upper-left: entity importance in the observation, upper-middle: final behavior performance after occluding each entity, upper-right: significancy test for the final behavior after occlusion, lower-left: joint importance in the action, lower-middle: final behavior performance after blocking each joint, lower-right: significancy test for the final behaviors after blocking.

performance bar in the bottom-middle figure proves the correctness of this hypothesis by showing that the amount of drop in performance after blocking `foot_left` and `foot` joints is substantial compared to the `standard` setting. For `leg` and `leg_left` the amount of drop in performance is close to each other. Both `leg` and `leg_left`'s importance scores explain this drop; however, the `leg_left`'s importance is low, although it was expected to be almost similar to `leg`'s score. One interesting thing is that the performance improves after blocking `thigh` and `thigh_left` joints. Although LRP gives a pretty low importance score to these two joints, it does not provide enough information to explain this performance improvement.

### 4.2.2 FETCHREACH-v1

First, we focus on analyzing the observation entity importance in the upper-row plots of Figure 3. The upper-left bar plot indicates that the most important entity is the `goal`. As expected, when occluding the `goal`, the performance drops significantly, as indicated in the upper-middle plot. After `goal`, `shoulder_pan` and `forearm_roll` entities have the highest scores. Their corresponding performance bars show a proportional amount of drop in performance after occlusion. We expect the `wrist_roll` joint to receive a relatively low importance score based on its performance bar. Although the `wrist_roll` joint seemed to be critical according to its importance score, if we look at the heat map of the `FetchReach-v1` in Figure 1, the action applied to this joint gave a relatively low score to the `goal`. Thus, we can conclude that this joint does not contribute to reaching the goal. That is why after occluding it, the performance did not change. The high score of `wrist_roll` entity is only because of the score given by the `wrist_roll` action. The importance score of the `upperarm_roll` entity can also be verified by its performance bar. However, since the `upperarm_roll` and `elbow_flex` performance bars are approximately equal, we expect a high score for the `elbow_flex`, unlike its current importance score. It remains `shoulder_lift` and `wrist_flex` entities that, as clear from their performance bars, the amount of drop in their performance can imply their importance score.

The joint importance in the action space evaluation is reflected in the lower-row plots of Figure 3. As indicated on the lower-left bar plot, the most critical joints to the actions are `elbow_flex`,

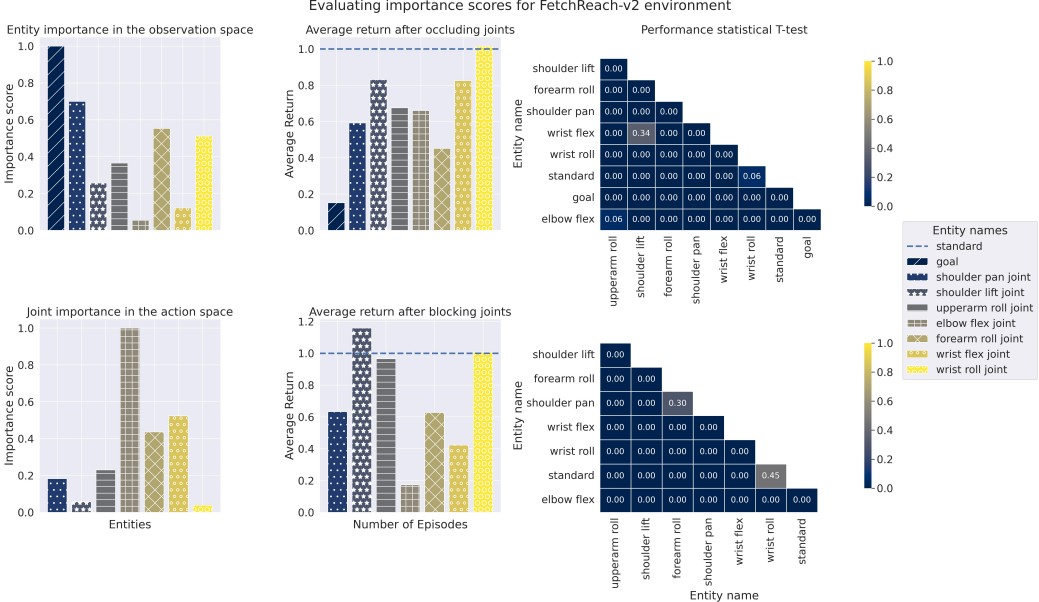

Figure 3: Evaluating explanation for the `FetchReach-v1`. Upper-left: entity importance in the observation, upper-middle: final behavior performance after occluding each entity, upper-right: significancy test for the final behavior after occlusion, lower-left: joint importance in the action, lower-middle: final behavior performance after blocking each joint, lower-right: significancy test for the final behaviors after blocking.

`wrist_flex`, and `forearm_roll` joints, respectively, and are highly strategic for reaching the goal. The noticeable drop in the performance after blocking these three joints, shown in their performance bar in the lower-middle plot, implies their importance. As discussed, the `wrist_roll` joint does not contribute to reaching the goal. Therefore neither its occlusion nor its blockage affects the performance. For `upperarm_roll` and `shoulder_lift` joints, the drop in their performance bar compared to the `standard` setting can be correctly explained by their importance score. Nevertheless, the LRP fails to explain the performance improvement after the `shoulder_lift` joint's blockage. For `shoulder_pan` joint, we expect that LRP gives an importance score approximately equal to the `forearm_roll` joint because their performance is nearly the same.

## 5 CONCLUSION AND FUTURE WORK

This paper proposes a novel technique for interpreting the deep reinforcement learning algorithms in robotic domains using graph neural networks and Layer-wise Relevance Propagation. This method identifies the contribution of the robot's components to the decision-making process, allowing us to analyze the learned behavior. The experimental results prove that our method could successfully highlight the importance of each part of the robot to decision-making. Although the contribution scores given to some entities were unexpected, for some of them, their importance could be explained by referring to the original heat map, as we did for the `wrist_roll` in the `FetchReach-v1` environment. The LRP also fails to provide additional information about the performance improvement after blocking some joints. Knowing the contribution of every part of the robot to performing a task is paramount, especially in realizing how important each part is in every stage of learning a task. Furthermore, during a malfunction, it can give us an intuition of the severity of the damage and whether adaptation is possible or not. Even after the adaptation, it provides additional information about how the robot substitutes the malfunctioning part by utilizing another internal part, which can be achieved by comparing the contribution scores of the two parts before and after adaptation.

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

## A   COMPUTATION STEPS IN A GRAPH NEURAL NETWORK

The computation steps of a GN block is summarized in algorithm 2. These steps comprise of 3 main updates:

1. The first part relates to edge updates. The algorithm updates edge attributes in lines 2-4 using the $\phi^e$ function.

2. The second part is dedicated to updating node attributes (lines 5-9). In this part, first, we select the set of edges whose receiver node is node $i$ denoted by $E'_i$ (line 6). Then the edge features of the edges in $E'_i$ would be aggregated using $\rho^{e \rightarrow v}$ to be used in the update function $\phi^v$ to update attributes of node $i$ in line 8.

3. The last part is updating global attributes (lines 10-14). $V'$ and $E'$ denote sets of updated node and edge attributes, respectively. Firstly, the updated edge attributes are aggregated using $\rho^{e \rightarrow u}$ and called $\overline{e}'$. Secondly, the updated node attributes are aggregated using $\rho^{v \rightarrow u}$ and called $\overline{v}'$. Then these aggregated node and edge attributes are used in updating global attributes using $\phi^u$.

The order of steps in algorithm 2 is irrelevant. One can change this order to update global attributes, per-node attributes, then per-edge attributes.

---

**Algorithm 2** Steps of computation in a GNN block

---

1: **function** GRAPH NEURAL NETWORK$(E, V, \mathbf{u})$
2:     **for** $k \in \{1 \dots N^e\}$ **do**
3:         $\mathbf{e}'_k \leftarrow \phi^e(\mathbf{e}_k, \mathbf{v}_{r_k}, \mathbf{v}_{s_k}, \mathbf{u})$                   $\triangleright$ Compute updated edge attributes
4:     **end for**
5:     **for** $i \in \{1 \dots N^v\}$ **do**
6:         **let** $E'_i = \{(\mathbf{e}'_k, r_k, s_k)\}_{r_k=i, k=1:N^e}$
7:         $\bar{\mathbf{e}}'_i = \rho^{e \to v}(E'_i)$                        $\triangleright$ Aggregate edge attributes per node
8:         $\mathbf{v}'_i = \phi^v(\bar{\mathbf{e}}'_i, \mathbf{v}_i, \mathbf{u})$                $\triangleright$ Compute updated node attributes
9:     **end for**
10:    **let** $V' = \{\mathbf{v}'_i\}_{i=1}^{N^v}$
11:    **let** $E' = \{(\mathbf{e}'_k, r_k, s_k)\}_{k=1}^{N^e}$
12:    $\bar{\mathbf{e}}' = \rho^{e \to u}(E')$                       $\triangleright$ Aggregate edge attributes globally
13:    $\bar{\mathbf{v}}' = \rho^{v \to u}(V')$                      $\triangleright$ Aggregate node attributes globally
14:    $\mathbf{u}' = \phi^u(\bar{\mathbf{e}}', \bar{\mathbf{v}}', \mathbf{u})$               $\triangleright$ Compute updated global attribute
15:    **return** $(E', V', \mathbf{u}')$
16: **end function**

---

## B   LAYER-WISE RELEVANCE PROPAGATION METHOD

The LRP method is explained according to the work by Bach et al. (2015). the relevance scores from higher layers are introduced as messages sent from those layers. Therefore, the relevance of a neuron $i$ at layer $l$ (except the last layer) is computed as follows:

$$R_i^{(l)} = \sum_{k:\ i \text{ is input for neuron } k} R_{i \leftarrow k}^{(l, l+1)} \tag{8}$$

where $R_{i \leftarrow k}^{(l, l+1)}$ shows the relevance coming from neuron $k$ in layer $l + 1$ to neuron $i$ in layer $l$. The relevance of the last layer is defined as the classification score $f(x)$. Equation 8 checks the sum of relevance scores with respect to the output neurons from the input neuron. We can consider the other way and check the sum of relevance scores of the input neurons for the output neuron:

$$R_k^{(l+1)} = \sum_{i:\ i \text{ is input for neuron } k} R_{i \leftarrow k}^{(l, l+1)} \tag{9}$$

Equations 8 and 9 are the main constrains of defining LRP. Multi-layer networks are commonly built as a set of interconnected neurons organized layer-wise. We denote neurons from layer $l$ by $x_i$ and neurons form layer $l + 1$ by $x_j$. In the same manner, the summation over all neurons of layers $l$ and $l + 1$ are denoted by $\sum_i$ and $\sum_j$ respectively. A common mapping from one layer to the next one consists of a linear projection followed by a non-linear function:

$$z_{ij} = x_i w_{ij} \ , \tag{10}$$

$$z_j = \sum_i z_{ij} + b_j \ , \tag{11}$$

$$x_j = g(z_j) \tag{12}$$

where $w_{ij}$ is the weight connecting neuron $x_i$ to neuron $x_j$, $b_j$ is the bias term, and $g$ is a non-linear activation function. Common non-linear functions can be Rectified Linear Unit (ReLU) or hyperbolic tangent (tanh). One possible choice of relevance decomposition for messages from layer $j$ to layer $i$ is as follows:

$$R_{i \leftarrow j}^{(l, l+1)} = \frac{z_{ij}}{z_j} . R_j^{(l+1)} \tag{13}$$

This type of formalization guarantees the conservation properties of equation 2. One drawback of the equation 13 is that for small values $z_j$, $R_{i \leftarrow j}$ can take unbounded values. Two solutions provided to overcome this drawback are $\varepsilon$-stabilizer and $\alpha\beta$-stabilizer.

For the $\varepsilon$-stabilizer, let $\varepsilon \geq 0$, then the relevance scores would be as follows:

$$R_{i \leftarrow j}^{(l, l+1)} = \begin{cases} \frac{z_{ij}}{z_j + \varepsilon} . R_j^{(l+1)} & z_j \geq 0 \\ \frac{z_{ij}}{z_j - \varepsilon} . R_j^{(l+1)} & z_j < 0 \end{cases} \tag{14}$$

One problem with this method is that the relevance can be fully absorbed if the stabilizer $\varepsilon$ becomes very large. For this case, we use the alternative $\alpha\beta$-stabilizer, which treats negative and positive pre-activations separately. Let $z_j^+ = \sum_i z_{ij}^+ + b_j^+$ and $z_j^- = \sum_i z_{ij}^- + b_j^-$ be negative and positive part of pre-activation respectively, where "+" and "−" denote the negative and positive values of $z_{ij}$ and $b_j$. The relevance scores are then calculated as follows:

$$R_{i \leftarrow j}^{(l,l+1)} = R_j^{(l+1)} \left( \alpha . \frac{z_{ij}^+}{z_j^+} + \beta . \frac{z_{ij}^-}{z_j^-} \right). \tag{15}$$

where $\alpha + \beta = 1$. This method can also control the importance of positive and negative evidence by changing the values of $\alpha$ and $\beta$. The complete layer-wise relevance propagation procedure for neural networks is summarized in algorithm 3.

---

**Algorithm 3** Layer-wise relevance propagation for neural networks

---
1: **let** $R^{(L)} = f(x)$
2: **for** $l \in \{L - 1 \ldots 1\}$ **do**
3:      $R_{i \leftarrow j}^{(l,l+1)}$ as in equation 14 or 15
4:      $R_i^{(l)} = \sum_j R_{i \leftarrow j}^{(l,l+1)}$
5: **end for**
6: **return** $\forall d : R_d^{(l)}$

---

