# OpenReview forum: "Explainability of deep reinforcement learning algorithms in robotic domains by using Layer-wise Relevance Propagation"
_ICLR.cc/2023/Conference — Submitted to ICLR 2023_

### Official Review · Reviewer_xzSU · 2022-10-12

**Confidence:** 5
**Correctness:** 3
**Technical Novelty And Significance:** 2
**Empirical Novelty And Significance:** 2
**Recommendation:** 5

**Clarity, Quality, Novelty And Reproducibility:**

The overall presentation of the work is clear, though elements of the results need further explanation (as in the statistical significance tests discussed above).

The overall novelty of the work is not particularly striking, as the primary contribution is applying an existing technique to a new style of model (applying LRP to GNNs). While this presents potentially useful or insightful takeaways for new domains or applications, the domains presented in this work do not highlight the true power of LRP for GNNs, as the state-action spaces are too small to get a sense for the potential of this technique applied to a GNN.

The technique seems to be reproducible.

**Strength And Weaknesses:**

Strengths:
* Needs for explainability in DRL are well stated, and preliminaries/background descriptions of GNNs, LRP, and SAC are clearly stated.
* Ablations are conducted to empirically validated the feature importances that are identified by the proposed method, lending support to the validity of the proposed method.

Weaknesses:
* General weaknesses of saliency methods are not discussed or addressed in this work (e.g., [1]). For example, while an individual feature may have a high saliency score, applying causality or further meaning to that score is unjustified and unfounded speculation.
* Statistical tests are performed at alpha=0.05, but there does not appear to be any correction (e.g., Bonferroni correction) for the dozens of statistical tests that have been performed. It is possible that many of the results are not significant after applying a correction, but the true statistical values are not provided beyond 2 significant figures, so this cannot be confirmed. Furthermore, effect sizes should be reported if significance tests are going to be used to show whether or not the significant effect is meaningful, and to what extent it is meaningful.
* The LRP approach to post-hoc explanation for RL agents is only applied to small state spaces with very clear alignments between state-action mappings. Experiments with larger, less-structured state spaces would be more illuminating to show whether or not the discovered features are truly meaningful.
* There are no constraints on sparsity for the discovered features, resulting in several rows of the confusion matrices (Figure 1) having several high-scoring elements. Again, this does not suggest the method would scale usefully to larger domains.
* For both domains, removing certain elements of the action space is shown to improve performance, yet these joints are not identified as being exceptionally important by the proposed method.

[1] Adebayo, Julius, et al. "Sanity checks for saliency maps." Advances in neural information processing systems 31 (2018).

**Summary Of The Paper:**

The paper proposes a new approach to interpretation of deep reinforcement learning agents via layer wise relevance propagation (LRP) applied to graph neural networks used in RL. The proposed approach produces feature-importance scores for different input features to a GNN, allowing for explanations via saliency for final policies. Salience scores are computed by backpropagating from individual actions through the GNN for multiple episodes, producing scores that can then be normalized and compared. Scores are validated by ablation studies on individual features to determine their true importance.

**Summary Of The Review:**

While it is interesting to see saliency methods applied to GNNs and to deep reinforcement learning in new ways, the approach proposed in this work lacks sufficient novelty or innovation to justify a strong accept without very strong experimental evidence. Unfortunately, the experiments are also lacking, with possible statistical errors and speculative explanations for feature importance scores. Without an overhaul of the discussions to either remove or substantiate causal descriptions of the feature importance and improvements to the results reporting, the paper is not yet ready for acceptance.

---

> ### Author Response · Authors · 2022-11-17
> **Thank you for your helpful feedback!**
>
> Thank you for your comments that help us improve the paper and clear some concepts.
>
> This is unfortunate that we did not discuss the downsides of the saliency methods. Thank you for mentioning this. We
> will definitely point this out in the introduction section.
>
> In statistical T-test, our target is not to show that all the performances are significantly different in general.
> Therefore, performing a correction does not have any meaning here since the correction takes into account all the tests
> simultaneously; instead, we want to analyze each test separately. We want to show that for those entities or action
> elements with almost similar importance scores, their performance after occlusion or blockage should be similar (
> having the same amount of drop in performance). To show this similarity, we used statistical T-tests.
>
> The target of our experiments is robotic domains; therefore, all the state spaces are structured. Even in robotic
> environments with visual inputs, the state spaces are structured images. Furthermore, since our focus is on robots with
> sensory inputs, the state spaces would not be that large. Could you please elaborate more on your comment?
>
> There might be a slight misunderstanding. Figure 1 does not represent any confusion matrix. It shows the relevance
> scores given by each action element in the output of the policy to every component of the input observation. The
> resulting heat map reflects this output-input relevance.
>
> The improvement in performance after removing certain elements is part of our future study. To answer this question, we
> are working on visualizing output-input relevance scores during the training process in addition to the converged
> behavior. It gives us an intuition of those parts' contributions during the learning process and the converged behavior
> separately.
>
> ### Contribution and Novelty
>
> We tried to explain more about the contributions and novelties of our work in answers to other reviewers' comments. For
> the discussion on the novelties and contributions of our work, please refer to the **Novelty** and **Clarification on
> the contribution** section in answer to the reviewer "jwcH."

---

### Official Review · Reviewer_ig8z · 2022-10-23

**Confidence:** 3
**Correctness:** 3
**Technical Novelty And Significance:** 2
**Empirical Novelty And Significance:** 2
**Recommendation:** 5

**Clarity, Quality, Novelty And Reproducibility:**

I think the paper is sufficiently clear and well-structured. The paper and the supplementary materials should be enough to allow reproducibility.

**Strength And Weaknesses:**

- Strengths: topic very relevant for the scientific community and promising results.
- Weaknesses: dubious technical and methodological novelty and contribution.

**Summary Of The Paper:**

The authors propose to analyze the learned representation in a robotic setting by utilizing graph neural networks (GNNs). Using GNNs and Layer-wise Relevance Propagation (LRP), they represent the observations as an entity-relationship to allow us to interpret the learned policy. Finally, they evaluate their approach in two environments from MuJoCo.

**Summary Of The Review:**

The main contribution and novelty of the paper is a bit unclear: there are already works dealing with explainability of DRL techniques, and LRP has already been employed before to enhance explainability in DRL and GNNs. From this point of view, it's not clear to me what is the main technical and methodological contribution of this paper.

I think the clarity of the paper will be enhanced if a graphical representation of the overall proposal is included in a Figure. A general scheme of the proposed method, as well as more information regarding motivation and intuitions, will be very positive. In fact, it's not clear to me what an explanation exactly is in this particular scenario and how this can be effectively employed by a human user.

---

> ### Author Response · Authors · 2022-11-17
> **Thank you for your helpful feedback!**
>
> We appreciate the reviewer's feedback and comments on the clarification of our proposed method.
>
> ### Contributions
>
> LRP has previously been applied to explain DRL agents' decisions in environments with visual inputs such as Atari games.
> It has also been applied to GNNs to explain graph classification (not reinforcement learning). Our work focuses on
> robotic settings with sensory inputs (not Atari or other environments with visual inputs) as observations, highlighting
> the importance of each part w.r.t the decisions made in an episode. For more discussion on novelties and contributions
> of our work, please refer to the responses to reviewers "jwcH" and "wEee."
>
> ### Clarity
>
> We acknowledge the reviewer's point about the unclarity of the proposed method, and we will try to improve the
> discussion. The explanation in robotic environments is complex, and it's difficult to provide a high-level explanation
> of the behavior that is easily understandable by the human user. Furthermore, the continuous action space adds to this
> complexity because the actions at each time step do not have any proper meaning to the user. In this work, we tried to
> provide a building block for explaining the agent's behavior. A high-level sketch of the applications of this method to
> robotic tasks is discussed in the last paragraph of the Introduction section. For more discussion on our method's
> potential applications and future directions, please refer to the **Clarification** section of the response to
> reviewer "wEee."

---

### Official Review · Reviewer_jwcH · 2022-10-24

**Confidence:** 3
**Correctness:** 2
**Technical Novelty And Significance:** 2
**Empirical Novelty And Significance:** 1
**Recommendation:** 3

**Clarity, Quality, Novelty And Reproducibility:**

Clarity: The conceptual idea is clearly presented, however the details of the method and implementation remain vague, despite the abundance of formal definitions and equations. It remains hard (at least for me) to piece together what exactly is happening in which order. The overall idea is clear though.

Quality: Low to medium: the presented results are not ablated and not validated by further experiments. The obtained metrics are just reported and then discussed, but it remains unclear whether we should trust those numbers or this is just one of many outcomes we could have obtained.

Novelty: Limited. I would consider fresh the idea to trace back the contributions of weight back to the input space in robot domain, however the approach itself is, of course, not novel.

Reproducibility: High. The amount of detail provided is sufficient to recreate the proposed system, or at least something close to it in spirit.

**Strength And Weaknesses:**

Overall this work does not reach the level of evidence to support or address the claims and research questions postulated in the abstract and the introduction. While the aim of the work is explainability of DRL algorithms for robotics, the actual result only covers a mapping between actions and state space entities, but does not make the next step -- showing how this information can be exploited to explain DRL agent's decision-making or make the agent's behavior more explainable or predictable.

In my mind the provided mapping is a curious idea, and I like the idea of back-tracing contributions to joints and state space entities. However this offers only a small piece of the explainability puzzle and the manuscript does not provide any empirical (or theoretical) confirmation that having this tool at one's disposal would indeed help explain agent's behavior.

Did I understand correctly from description around Equation 4, that before the algorithm is trained, all of the function approximation networks are replaced by GNNs and the the standard DRL learning process is ran, but now we have GCN features instead of plain robot state vectors? If that is so, I would like to know how well does the resulting system perform after all these changes to the architecture.

I would be interested to discuss whether the use of GNN machinery is the easiest way to reach the mapping demonstrated in this work. Could we build the same mapping by using just state space saliency maps and the actions that were performed in those states? This is not a direct critique, but it seems that adding additional learning step (training a working GNN on top of the already existing learning task) makes the problem harder and the same outcome could have been achieved by analyzing the behavior statistic of the agent without the graph decomposition.

Result presented in sections 4.1.1 and 4.1.2 verbosely describe the observations presented in Figure 1, but provide little in addition to what we can already see on the Figure. Many of claims made in those sections are speculative, but are presented as facts (such as "This case highlights one of the reasons we selected the graph structure: not only do we take into account the effect of features of each entity on the decision-making process, but we also consider their position in the structure.")

I find that the manuscript lacks the mention of the final performance of the agent that was subsequently analyzed by the proposed method. There is a mention of "training to convergence", but is not specified at what level of performance that convergence happened.

To validate the applicability of the proposed method I would also need some demonstration of the proposed metric leading to a useful insight or some other kind of application.


**Summary Of The Paper:**

The paper describes an approach to transform state space into a graph representation and create an importance mapping between state entities and action entities. This can be seen as a tool in the explainability toolbox, however in my estimation the manuscript does not demonstration how this metric could be used in application contexts mentioned in the abstract (understanding of decision-making, recovery from faults). The main result is a mapping for two MuJoCo environments and verbal analysis of whether various portions of this mapping make sense.

**Summary Of The Review:**

I find this work lacking in many important aspects, mainly the finding only to a small per cent supporting the claims and aspirations of the paper. I am unable to judge the explanatory power of the proposed methods as no such validation is provided in the work: maybe this is an amazing metric, or maybe these number are just marginally useful statistics of a DRL policy -- at this point this remains unknown. My recommendation is to exclude this paper from the conference.

---

> ### Author Response · Authors · 2022-11-17
> **Thank you for your helpful feedback!**
>
> We appreciate your helpful reviews for improving the paper!
>
> ### Clarification on the contribution:
>
> This work forms the building block for explaining and predicting the robot's behavior when changing dynamics. The
> importance scores predict the robot's performance if something goes wrong in a specific part (
> as we did in section 4.2). The potential directions we are currently pursuing to elucidate the applications of our work
> are discussed in the **Clarification** part of the answer to reviewer "wEee."
>
> ### Clarification on the use of GNN:
>
> The method is the same as you discussed. GNNs generalize better than fully connected networks, which makes them harder
> to train and converge. Therefore, the average reward of the final behavior is lower than fully-connected networks in
> many MuJoCo environments. However, the performance is stable with a lower variance. In some environments, such as the
> original FetchReach-v1, GNNs perform much better than the fully-connected networks. We aimed to consider the position of
> each part relative to other parts in calculating relevance scores (for the reasons discussed in the section 4.1).
> However, we will also consider applying LRP to the fully-connected networks to see whether it would achieve the same
> results. The GNN is not trained on top of the learning task; it is actually the learning task. The function
> approximators here are GNNs instead of fully connected networks. So it does not add any additional learning steps.
>
> ### Discussion on the results
>
> We acknowledge the reviewer's point about the discussion of our results. It should be mentioned that this is an
> empirical paper, and we try to make conclusions based on our observations. We would definitely adopt a language that is
> less certain and close to empirical conclusion.
>
> The converged behavior means that the agent can perform properly in the environment, and its average reward is similar
> to the standard setting.
>
> The Spearman's ranking correlation coefficient, suggested by one of the reviewers, can bring a useful insight into the
> application of our method. This coefficient highlights the inverse correlation between the importance scores and
> performance. The closer it gets to -1, the stronger inverse correlation it shows. This coefficient for the FetchReach-v1
> action importance case was -0.92, which shows an almost perfect inverse correlation; for the observation importance, it
> was -0.57, which shows a strong inverse correlation. For the Walker2d-v2, the scores for action and observation
> importance are -0.6 and -0.75, respectively, which shows a very strong inverse correlation between the importance scores
> and performance bars.
>
> ### Clarity:
>
> We acknowledge that the proposed method might be vague. We will try to add more description to the methodology for
> clarification.
>
> ### Novelty:
>
> * To the best of our knowledge, only a few past works focused on explaining continuous action-space RL. Previously, most
>   XRL methods focused on an environment with a discrete action space, similar to classification.
> * To the best of our knowledge, this is the first work focused on ranking the robot's components based on their
>   contribution to the decision-making.
> * One of the essential parts of providing an explainability method is to propose an evaluation metric for the
>   explanations. XRL is a new field of study, and metrics for evaluating explanations are not well-defined. We provided
>   an evaluation strategy for measuring the accuracy of the explanations for dynamically changing environments.
> * To the best of our knowledge, all the previous XRL studies in robotics focused on environments with visual inputs. It
>   is the first work focusing on a robotic domain with sensory input and without any camera or visual observations of the
>   environment.
> * This work provides a foundation for explaining the **adaptation** in robotics, which is one of the major superiority
>   of RL over traditional control theory.

---

### Official Review · Reviewer_wEee · 2022-10-24

**Confidence:** 4
**Correctness:** 1
**Technical Novelty And Significance:** 2
**Empirical Novelty And Significance:** 2
**Recommendation:** 3

**Clarity, Quality, Novelty And Reproducibility:**

The motivation for this method is unclear.
In explainability, there is generally a use-case. An example use-case would be to allow a human to predict the next action or to predict the action if something were to change.
The last paragraph in Section 1 serves to motivate the work, but:
- The listed applications do not permit creating testable hypotheses that an explanation can help test.
- The use-case of "figure out how severe the damage is" cannot be done with this method: if the agent can solve task in either case (for example, stand up with hands or without them), then performance would not be affected; however, agent's specific actions would be greatly affected by presence of hands, so saliency methods would highlight them.
- For the use-case of "explain the adaptation process," what is the explanation? What about the adaptation process has been explained? (The same goes for "visualization for explaining the training process")
- This section lists aspects that would be useful to explain, but there is no connection between the explanation and making conclusions of the specified types. The mechanism for making the conclusions would then form the basis of tests performed later in the paper.

To make room for thorough evaluation, the explanation of LRP can be substantially reduced. It is sufficient to outline LRP at a high level.
Likewise, in Section 4, there are paragraphs that transcribe information that is already present within the plots (without making any conclusions from this information). These redundancies can be reduced to make more room.

Further Minor Comments:
- The plots are often far from the text focused on describing them. Figure 3, in particular, is two pages ahead of its text. If possible, placing figures closer to their text would help readability.
- "Saliency methods have proved to be successful in highlighting the most relevant pixels in image classification" has been brought into question. At the very least, some saliency methods have been found to be weight-agnostic.
- It is insufficient to refer to results on LRP for graph classification to support the use of LRP in this use-case.
- The definition of the policy is odd. Suggests that probability of an action in a given state can vary over time.
- top of page 7: "(bottom bar plot in Figure 2" -> "(bottom bar plot in Figure 2)"
- 4.1.2: Referring ot FetchReach-v1 in Figure 1 while Figure 1 says v2

**Strength And Weaknesses:**

This work is missing qualitative evaluations of the proposed method.
Visualizations of the explanations should be secondary to evaluations.
Guesses are made as to how the agent functions / why the explanation is of a certain form (e.g., 4.1.1), but these need to be tested, at the very least.
Some of these guesses also raise concerns. For example, the authors note that "the vicinity of the two entities in the robot is the cause of the high score". If this is the case, this does not seem to be a desirable property.
This same guess is mentioned in 4.1.2. However, an explanation method that assigns scores based on known connectivity information does not provide more information to a potential user.
These same guesses do not fully match the examples shown. Continuing with example: This pattern does not hold for other adjacent entities (e.g., if this is reason for leg_left and thigh_left, then why does this not happen for leg and thigh?).

Section 4.2 attempts to "validate the correctness of the hypotheses mentioned in Section 4.1". This is a great goal, but the approach does not achieve this goal.
Most importantly, all explanations are for a different, separately trained agent. Since the proposed approach is seeking to explain a policy/agent as opposed to the task, hypotheses from 4.1 cannot be validated using a separate agent.
In addition, the results do not support the hypotheses made. This is unsurprising, though, since the explanation method used in 4.1 differs from those used in 4.2 and are for different policies (thus, different information about different behavior).

The results are also not in a qualitative form, but comments made after-the-fact based on plots. It is important to generate a hypothesis and then test it to see how often the proposed method is useful.
(The ability to draw matching conclusions from two different explanation formats is of limited use. If one is supposed to be predictive of the other, the ability to predict should be shown.)
For example, if the scores are meant to help predict the effect of occluding a feature, then the ranking correlation between the scores and occlusion outcomes can be computed.

Some comments on methodology:
- If LRP is supposed to decompose the contribution across a layer (while keeping the same total relevance within each layer), then what is the motivation for approach #2 in Section 3?
- Occluding features does not test which feature presence is important for success, but which features the agent seeks to observe. Thus, it cannot be used to test for entity importance.
- There is no clear statement about being a local or a global explanation. LRP is a local method, yet this work is closer to a global explanation via averaging of local explanations. This approach will yield odd results in more complex domains (e.g., one where two separate behaviors are executed).

**Summary Of The Paper:**

This work proposes a method for explaining DRL behavior in a robotics setting by using a graph of robot components as input.
This input choice enables LRP to identify the importance of different components. By combining values across states or actions, different explanations are generated.

**Summary Of The Review:**

This work presents a method for generating explanations, but no way to draw conclusions from these explanations.
Relatedly, the method is not evaluated, so its usefulness has not been demonstrated.

---

> ### Author Response · Authors · 2022-11-17
> **Thank you for your valuable feedback!**
>
> We appreciate your insightful reviews that help elucidate the concept.
>
> The heat maps generated could not be tested as they are. Therefore, we turned those heat maps to bar plots, showing the
> importance of every entity or action element. The purpose of the heat map is to show the individual dependence of every
> action element on each observation entity and the goal of the environment. It creates a diagonal pattern in the heatmap,
> showing the high relevance between the state of an entity and the action applied to it. The pattern happened for the
> left leg but not the right leg because, according to the simulation videos, the agent mainly uses its left leg. We
> should have pointed that out in the main text.
>
> Thank you for pointing out the ambiguity in our evaluations. The explanations, agents, and learned policy in sections
> 4.1 and 4.2 are similar. We have only changed the heat map representation into importance bar plots (with minor changes)
> to reflect the importance of every observation entity and action element. The importance scores try to predict the drop
> in performance after occluding the entity's features in the observation space or blocking the corresponding joint in the
> action space. We used agents with fully-connected networks as function approximators to generate the performance bars (
> which are not the target of explanation and are only for evaluation purposes). In addition, the results almost perfectly
> matched our expectations.
>
> Thank you for this great suggestion. We calculated Spearman's ranking correlation coefficient for both environments for
> the importance of action and observation. This coefficient for the FetchReach-v1 action importance case was -0.92, which
> shows an almost perfect inverse correlation; for the observation importance, it was -0.57, which shows a strong inverse
> correlation. For the Walker2d-v2, the scores for action and observation importance are -0.6 and -0.75, respectively,
> which shows a very strong inverse correlation between the importance scores and performance bars.
>
> ### Methodology:
>
> * We hypothesized that LRP provides a one-to-one correspondence between action elements and observation entities.
>   Therefore, averaging the scores for each action element across different entities provides the importance of the
>   action element.
> * By entity importance, we exactly meant which features the agent seeks to observe (and are important for the policy).
> * Since the action space is continuous, applying LRP to single decisions does not give any helpful insight. However, in
>   every step of decision-making, LRP can highlight the contribution of each part to the decision. Hence, averaging the
>   contributions throughout the episode produces the overall contribution of each part to reaching the goal.
>
> ### Clarification:
>
> * The purpose of this explanation is to predict the agent's behavior when the dynamics change. As discussed in the
>   experiments, if part A is damaged, we expect an amount of drop in the performance proportional to the importance of
>   part A.
> * Here, we want to point out a list of methods that we are currently working on to test the application of our method:
>     * "Visualization of the training process": For this purpose, we halt the training process every a number of
>       iterations and calculate the relevance scores. The scores change during training time. By visualizing the change
>       in scores across time steps, we can understand what parts are more important in which training step. We can then
>       evaluate those claims using the same method discussed in our paper. This way, we take into account the
>       contribution of each part both to learning and after convergence.
>     * "Figure out how severe the damage is": This was evaluated and tested in the current work. It shows that the amount
>       of drop in performance is predictable by the LRP scores. Furthermore, one can decide whether a policy trained in a
>       normal situation is transferable to the new setting. It is transferable if the importance score of the damaged
>       part is low.
>     * "Explain the adaptation process": Comparing the LRP scores in the normal situation, after a malfunction, and after
>       adaptation to the malfunction explains how the agent adapted to the new dynamics. For example, in the walker-2d
>       environment, if the left leg breaks, the agent should adapt to use its right leg. Therefore, one can figure out
>       this change by comparing their importance score before and after adaptation.
> * In the "stand-up" example, although the agent can stand up without hands, the absence of hands makes the training
>   process harder and slower. However, what if the legs were affected?

---

### Decision · Program_Chairs · 2023-01-20

**Decision:**

Reject

**Justification For Why Not Higher Score:**

Limited novelty and poor execution.

**Justification For Why Not Lower Score:**

N/A

**Metareview: Summary, Strengths And Weaknesses:**

The paper proposes to use graph neural networks and layer-wise relevance propagation for explaining deep reinforcement learning in robotic settings.  As pointed out by multiple reviewers and acknowledged by the authors "LRP has previously been applied to explain DRL agents' decisions in environments with visual inputs such as Atari games. It has also been applied to GNNs to explain graph classification (not reinforcement learning)", the new contribution here is to apply this technique to a new domain, i.e., robotic settings. The authors argued that the new setting brings a few differences, such as continuous action space and sensory input. It is unclear though what changes the authors have to adopt to enable LRP in the new settings, and therefore hard to gauge if the learning will be of interest to the community. Reviewers also raised concern that the work does not support or address the claims the research questions postulated in the abstract and the introduction. It is also hard to draw conclusion from the different explanation methods used in the two different environments.